# Stratification of Length of Stay Prediction following Surgical Cytoreduction in Advanced High-Grade Serous Ovarian Cancer Patients Using Artificial Intelligence; the Leeds L-AI-OS Score

Alexandros Laios [1,*], Daniel Lucas Dantas De Freitas [2], Gwendolyn Saalmink [1], Yong Sheng Tan [1],
Racheal Johnson [1], Albina Zubayraeva [1], Sarika Munot [1], Richard Hutson [1], Amudha Thangavelu [1],
Tim Broadhead [1], David Nugent [1], Evangelos Kalampokis [3,4], Kassio Michell Gomes de Lima [2],
Georgios Theophilou [1] and Diederick De Jong [1]

[1] ESGO Centre of Excellence for Ovarian Cancer Surgery, Department of Gynaecological Oncology,
   St James's University Hospital, Leeds Teaching Hospitals, Leeds LS9 7TF, UK
[2] Department of Chemistry, Federal University of Rio Grande do Norte, Natal 59078-970, Brazil
[3] Information Systems Lab., Department of Business Administration, University of Macedonia,
   54636 Thessaloniki, Greece
[4] Center for Research & Technology HELLAS (CERTH), 6th km Charilaou-Thermi Rd.,
   57001 Thessaloniki, Greece
* Correspondence: a.laios@nhs.net

**Abstract:** (1) Background: Length of stay (LOS) has been suggested as a marker of the effectiveness of short-term care. Artificial Intelligence (AI) technologies could help monitor hospital stays. We developed an AI-based novel predictive LOS score for advanced-stage high-grade serous ovarian cancer (HGSOC) patients following cytoreductive surgery and refined factors significantly affecting LOS. (2) Methods: Machine learning and deep learning methods using artificial neural networks (ANN) were used together with conventional logistic regression to predict continuous and binary LOS outcomes for HGSOC patients. The models were evaluated in a post-hoc internal validation set and a Graphical User Interface (GUI) was developed to demonstrate the clinical feasibility of sophisticated LOS predictions. (3) Results: For binary LOS predictions at differential time points, the accuracy ranged between 70–98%. Feature selection identified surgical complexity, pre-surgery albumin, blood loss, operative time, bowel resection with stoma formation, and severe postoperative complications (CD3–5) as independent LOS predictors. For the GUI numerical LOS score, the ANN model was a good estimator for the standard deviation of the LOS distribution by ± two days. (4) Conclusions: We demonstrated the development and application of both quantitative and qualitative AI models to predict LOS in advanced-stage EOC patients following their cytoreduction. Accurate identification of potentially modifiable factors delaying hospital discharge can further inform services performing root cause analysis of LOS.

**Keywords:** machine learning; deep learning; artificial intelligence; surgical cytoreduction; epithelial ovarian cancer; length of stay; graphical user interface



## 1. Introduction

Cancer of the fallopian tube, ovary, or peritoneum (EOC) is the leading cause of death from gynecological malignancy in the western world [1]. Over 70% of women diagnosed with EOC have advanced disease at presentation (FIGO stage 3–4) [1]. High-grade serous ovarian cancer (HGSOC), yet the most prevalent, is now recognized as a single clinical entity. The treatment includes a combination of cytoreductive surgery and platinum-based chemotherapy. The surgery aims at maximal cytoreduction of all visible disease, ideally reaching a total macroscopic tumor clearance. When the cancer is at an advanced stage, surgery can be extensive resulting in prolonged hospitalizations.

Length of stay (LOS) is a measurable outcome that can be used as a benchmark of short-term surgical care. Through the introduction of the Enhanced Recovery after Surgery (ERAS) pathway, an effort has been made to shorten the LOS for patients following major surgery, whilst still assuring that they receive effective treatment and high-quality care [2]. The PROFAST trial focused exclusively on EOC and provided much-needed randomized evidence supporting surgical quality improvement in ERAS implementation compared with conventional management [3].

The accepted "ideal" LOS resulting from optimal major surgery is deemed to be five days or less [4]. Excessive or prolonged LOS has also been suggested as a marker of the effectiveness of care, as it may increase medical resource utilization, overall cost, readmissions, and short-term mortality [5]. Proposed definitions of prolonged LOS include time spent in the hospital beyond the median or the 90th percentile [6]. Identification of modifiable risk factors at admission, to predict LOS, could lead to appropriately targeted interventions. In our institution, the average LOS following major EOC surgery ranges between five and seven days. Inpatient stays longer than seven days are associated with a higher risk of post-discharge adverse outcomes and complications compared to short stays (less than seven days) regardless of admission causes [7].

In EOC, the factors affecting LOS following cytoreductive surgery are incompletely characterized. Most of the knowledge comes from the surgical oncology literature [8,9]. Such factors may include increasing age or frailty, low albumin, high comorbidity scores, and blood transfusion [6,10]. Optimal stoma care and severe surgical site infections (SSIs) could be responsible for prolonged LOS [11–13]. Risk-prediction algorithms for severe postoperative complications after ovarian cytoreductive surgery have been proposed [14]. Models to improve performance in predicting LOS have been additionally explored [15].

Modern public health studies use data mining technologies to explore the complex risk factors of diseases. Such sophisticated methods through low-cost computational methods represent significant advances in an era of stringent health economics. Machine learning (ML) and deep learning (DL) approaches to predict new data from identified patterns have been applied in a variety of hospital settings [16]. An accurate prediction can sometimes be difficult with conventional statistics because patient characteristics show a multidimensional and non-linear relationship. We previously employed ML algorithms to improve the prediction accuracy of complete cytoreduction in advanced HGSOC patients [17]. In addition, we highlighted the importance of feature selection for accurate 2-year prognosis estimation in the same population by use of ML [18]. The usefulness of ML as a prognostic tool in the ovarian cancer environment has been previously demonstrated [18]. Machine Learning methods could help monitor hospital stays to improve standards of care. We hypothesized that some of the factors affecting LOS are not endogenous, hence potentially modifiable. We sought to improve the accuracy of predicting LOS in advanced-stage HGSOC patients undergoing cytoreductive surgery using ML/DL algorithms. Equally, we aimed to develop a DL-based novel predictive LOS score and refine factors significantly affecting LOS. The primary outcome was the prediction accuracy of several ML methods, based on a set of performance metrics for differential LOS time points, and the investigation of feature associations. The secondary outcome was the development and generation of DL-driven real-time predictions in a post-hoc sub-cohort of HGSOC patients by treating LOS as a continuous variable.

## 2. Materials and Methods

Prospectively registered data from consecutive patients diagnosed with histologically proven diagnosis of advanced stage HGSOC, who underwent elective cytoreductive surgery from January 2014 to December 2019 at St James's University Hospital, Leeds by a certified Gynaecologic Oncology Surgeon was analyzed. Our tertiary center has been recently accredited by the European Society of Gynaecologic Oncology as a Centre of Excellence for ovarian cancer surgery. The patients were discussed at the central multidisciplinary team (MDT) meeting and were prospectively recorded in the hospital-wide

Patient Pathway Manager (PPM) electronic database. The study was approved by the Institutional Review Board (MO20/133163/18.06.20) and performed according to the standards outlined in the Declaration of Helsinki. The MDT criteria for the decision and timing of surgery have been previously reported [19]. Only HGSOC patients with at least one pre-treatment CA125 were included in the study. Patients aged <18 years, as well as those with progressive disease or recurrent disease undergoing secondary cytoreduction, were excluded. An Enhanced Recovery after Surgery (ERAS) pathway was implemented at our center in 2015. Facilities have been described in detail elsewhere [19]. Postoperative critical care unit (CCU) admission was electively booked for high-risk HGSOC patients, who were scheduled to undergo complex major surgery, including multi-visceral resections or undertook preoperative cardiopulmonary exercise (CPEX) fitness testing [20]. The training cohort consisted of consecutive patients from January 2014 to December 2017 (ERAS implementation and transition). Consecutive patients from January 2018 to December 2019 were included in the post-hoc validation cohort from the same institutional database using the same criteria as the derived cohort (ERAS evaluation).

Predictive variables were selected a priori from the hospital database. Demographic characteristics included age, year of diagnosis, body mass index (BMI), pre-operative co-morbidities using the Charlson Co-morbidity Index (CCI), Eastern Cooperative Oncology Group (ECOG) performance status (PS), preoperative albumin and timing of surgery (primary or interval). Intraoperative characteristics included operative time, surgical complexity score (SCS), disease score (DS), residual disease (RD), and estimated blood loss. These variables are widely available at tertiary centers and shown to be independent predictors of postoperative morbidity and mortality in EOC patients [14]. The postoperative complications were recorded according to the Clavien-Dindo classification (grade 3–5) (CD3–5) [21]. Only severe SSIs were recorded. The CCI was categorized as 0, 1, or 2; with higher scores indicating greater co-morbidity [22]. The SCS was assigned based on the Aletti classification as low, intermediate, and high [23]. Outcomes also included ideal LOS and prolonged LOS. Patients experiencing the optimal (ideal) stay of <five days were identified, as were those with a prolonged LOS > five days or >90th centile [24].

Descriptive statistics were displayed by frequency and percentages for binary and categorical variables and by means and standard deviations (SD) or medians (with lower or upper quartiles) for continuous variables. The Chi-square test was performed for categorical variables and Fischer's exact tests were used for binary variables. Conventional linear logistic regression (LR) was used as a baseline.

## 2.1. Feature Selection

Feature selection aimed to identify the smallest group of independent predictive variables with the highest association to the dependent variable to ensure the best performance and minimize over-fitting. All variables were evaluated individually, one by one against the continuous vector of LOS classification responses. Despite several feature selection methods serving the purpose, we selected this simple methodology objectively verifying the features and their respective values representing the highest correlation and $p$-value < 0.05 when compared to the continuous values of the variable of interest (LOS). All the variables that obtained significant correlation ($p$-value < 0.05) were included in the model to facilitate the understanding of the model construction, allowing feature reduction without the use of complex selection algorithms, especially when dealing with poorly correlated clinical variables, which could make the application difficult.

To optimize the latent collinearity and avert variable over-fitting, a correlation analysis was performed by assessing the determination coefficient ($R^2$) and the root mean square error of cross-validation (RMSEVC). A poor correlation between variables was observed if the $R^2$ was low or the RMSEVC was elevated. Variables with a threshold of >0.6 were eliminated to determine the input variables for the final model. The variables selected for use were the following: age; SCS; Disease Score (DS); Albumin Level; Estimated blood loss (EBL); Operative Time (OT); Bowel Resection; CCU Admission, and Clavien-Dindo

complications (3–5). These were variables readily available in the electronic health records as shown in Table 1:

**Table 1.** Table of all clinical, readily available variables (selected, *p*-value < 0.05, and unselected, *p*-value > 0.05) initially interrogated for the LOS prediction. Only variables with *p*-value < 0.05 were employed in both the calibration and classification process, and finally in the construction of the proposed prediction score.

| Variable | Type | *p*-Value |
|:---:|:---:|:---:|
| Age | Numerical | *0.049* |
| BMI | Numerical | 0.317 |
| Performance Status | Numerical | 0.293 |
| CCI | Numerical | 0.985 |
| Type of Surgery | Categorical | 0.124 |
| SCS | Numerical | *0.000* |
| Disease Score | Categorical | *0.002* |
| CA 125 | Numerical | 0.458 |
| Albumin | Numerical | *0.001* |
| EBL | Numerical | *0.000* |
| Operative Time | Numerical | *0.000* |
| Bowel Resection | Categorical | *0.000* |
| Residual | Numerical | 0.363 |
| R0 | Categorical | 0.262 |
| CCU Admission | Categorical | *0.000* |
| Clavien-Dindo complications | Categorical | *0.000* |

*2.2. Model Development*

The model development followed a standard data science approach. The raw clinical data were uploaded on the MATLAB environment version 8.4 (R2014b) (MathWorks Inc., Natick, MA, USA) for pre-processing and subsequent multivariate analysis. The Classification Toolbox (for MATLAB) was also used to build supervised classification models [25]. The PLS toolbox version 7.9.3 (Eigenvector Research, Inc., Manson, WA, USA) was also used to build calibration models. One patient with missing information was discarded from the dataset. Data normalization was carried out to avoid bias towards certain variables.

For the construction of classification models, the categorical variables were transformed into binary dummy variables and were labeled as positive or negative according to the classification problem. The LOS values for the total dataset were evaluated through histograms to assess the level of normality of these data by considering an average value with up to three standard deviations (99.7%), with a limit value of LOS equal to 24 days, within the normal Gaussian curve. Thus, eight samples were taken from the total set. The training and test groups were constructed using the Kennard-Stone uniform sample selection algorithm, widely used in the literature (calibration to prediction ratio, 70%:30%). The training group was used to build the initial model. With the model built, the training group was evaluated. The validation set was built from 50% of the samples, using the Venetian cross-validation (CV) Splits method. In this method, the dataset was randomly split into five, almost equal sub-samples; each one was used as a test set in one of five different feature selection processes using the remaining groups as a training set. This set served to optimize some parameters and to make an estimate of the prediction for those samples that were not used in the construction of the model. The test group consisted of those samples that were never used in either the training set or the cross-validation (CV) set. These were new samples tested by the best-fitted final model and their hit-and-miss values were extremely important as they demonstrated how the model reacted to new samples added. Lastly, only the features extracted in all five feature selection processes were selected as most relevant for the subsequent analysis steps. This scalable strategy enabled us to feature the regression models based on the reduced set of relevant variables. For the linear regression models, given the right-skewed distribution of all LOS measurements, the

dependent variable (i.e., LOS) was log-transformed. For model development, the selection of predictor variables was based on those variables that passed the multicollinearity test (variance inflation factor < 2.0). For the non-parametric models, all features were used and the algorithm itself determined which of those remained in the models. All model parameters were determined based on the training data set.

### 2.3. Multivariate Analysis

State-of-the-art supervised classifiers, suitable for the type and size of the dataset were tested in a multivariate fashion. The algorithms were chosen because of their relative simplicity for interpretation and through our testing of several ML algorithms during model development. The algorithms included LR methods; support vector machine (SVM) and artificial neural networks (ANN) in backpropagation (BPN) configuration.

The SVM-based classification is powerful, robust, and widely used in various knowledge domains. The model performs a non-linear data transformation to maximize the margins that separate the samples from different groups. The SVM classification rule is obtained when the following equation is used [26]:

$$f(x) = sign(\sum_{i=1}^{N_{SV}} \alpha_i y_i k(x_i, z_j) + b \tag{1}$$

where the value of $N_{SV}$ is the number of support vectors. $\alpha_i$ is the Lagrange multiplier operator, and $y_i$ is the class membership. $k(x_i, z_j)$ is the kernel function and $b$ is the bias parameter.

The Artificial Neural Network (ANN) was of the backpropagation (BPN) type. This algorithm works with learning layers of different sizes and configurations. Indeed, it is a type of DL because it "learns" and uses the artificial neurons to calculate the input and the final output of the model, estimating an error for each simulation, performing simulations, and adjusting their weights to ultimately obtain the smallest possible error. This is a strategy not routinely used by common ML algorithms, such as SVM, which has been used in this work and has proved to be highly robust in prediction problems. Unlike other algorithms, BPN controls errors by resubmitting flawed solutions to the initial neurons, allowing the backpropagation method to improve in the next iteration. These ML/DL paradigms have been comprehensively described in our previous work [27]. Clinicians are mostly familiar with regressions, odds ratios, and hazard ratios. Hence conventional logistic regression (LR) was used as a benchmark.

### 2.4. Model Performance

The performance of the different qualitative and quantitative models was evaluated differently, according to their nature. The classification models were evaluated through well-known figures of merit such as accuracy, sensitivity, specificity, F-score, and G-score, considering the results of the external test set. Accuracy measures the proportion of samples that were correctly identified in their respective groups considering the number of true and false negatives. Sensitivity measures the proportion of samples considered to be positive samples that were correctly identified. Specificity, on the other hand, measures the proportion of samples from the negative group that was correctly identified. The F-score and G-score figures of merit measure the performance of the models built considering the unbalanced dataset and without considering the size of the classes, respectively.

For the evaluation of quantitative models, figures of merit widely known as the Root Mean Square Error (RMSE), and the Bias value for the calibration, cross-validation, and prediction sets were used. The statistical quality parameters for the models were calculated as follows [28]:

$$Accuracy\ (AC) = \left( \frac{TP + TN}{TP + FP + TN + FN} \right) * 100 \tag{2}$$

$$Sensitivity\ (SENS) = \left(\frac{TP}{TP + FN}\right) * 100 \tag{3}$$

$$Specificity\ (SPEC) = \left(\frac{TN}{TN + FP}\right) * 100 \tag{4}$$

$$F - Score = \left(\frac{2 * SENS * +SPEC}{SENS + SPEC}\right) \tag{5}$$

$$G - Score = \sqrt{SENS * SPEC} \tag{6}$$

$$RMSE = \sqrt{\frac{1}{n}\sum_{i=1}^{n}(y_i - \hat{y}_i)^2} \tag{7}$$

*2.5. Development of the LEEDS L-AI-OS Score*

To generate real-time predictions, numerous LOS predictions at different model probability output thresholds were performed in the training and test sets. We employed the ANN (BPN) model in calibration mode to turn discrete values into continuous values. Nine features were selected as a threshold, offering the best compromise of computing time required and possible information loss. A range of values, including minimum and maximum, were set for each feature continuous and categorical as below:

Age–Continuous: 41–90 (years)
SCS–Continuous: 2–11
DS–Categorical: 1–3
ALB–Continuous: 27–49 (g/dL)
EBL–Continuous: 100–4000 (mL)
OT–Continuous: 65–480
Bowel Resection with Stoma–Categorical: 0–1 (no–yes)
CCU–Categorical: 0–1 (no–yes)
CD3–5–Categorical: 0–1 (no–yes)

A software tool (Calibration Tools Hall, UFRN, Natal, Brazil, upon request) was developed by using the ANN (BPN) regression algorithm based on the above-selected features. The algorithm fine-tunes the weights of a neural network based on the error rate obtained from the preceding iteration. Therefore, the error rates become reduced rendering the model more reliable.

All values were numerically entered or selected, resulting in the Graphical User Interface GUI) resetting to the closest minimum/maximum value in one or all input 9 variables. The 'Predict LOS' button was used to estimate the length of stay as a result. The EQUATOR Guideline for reporting ML predictive models and the STROBE statement for reporting observational studies were followed for the development of the predictive models [29,30].

## 3. Results

A flow chart of the study is illustrated in Figure 1. Descriptive cohort statistics are shown in Table 2. The cohort has been previously described [27]. A total of 201 (two samples removed) samples and 84 samples (three samples were removed from the calibration and post-hoc test sets, respectively) were used for the analysis. The mean and median LOS was 6.0 and 5.0 days (IQR 3–24), respectively. Histograms of the LOS distribution across the HGSOC cohort are shown in Figure 2. The rate of ideal LOS continuously improved for every subsequent year from 32% in 2016 to 73.5% in 2019 despite increasing mean SCS, reflecting the efficiency of the ERAS pathway.

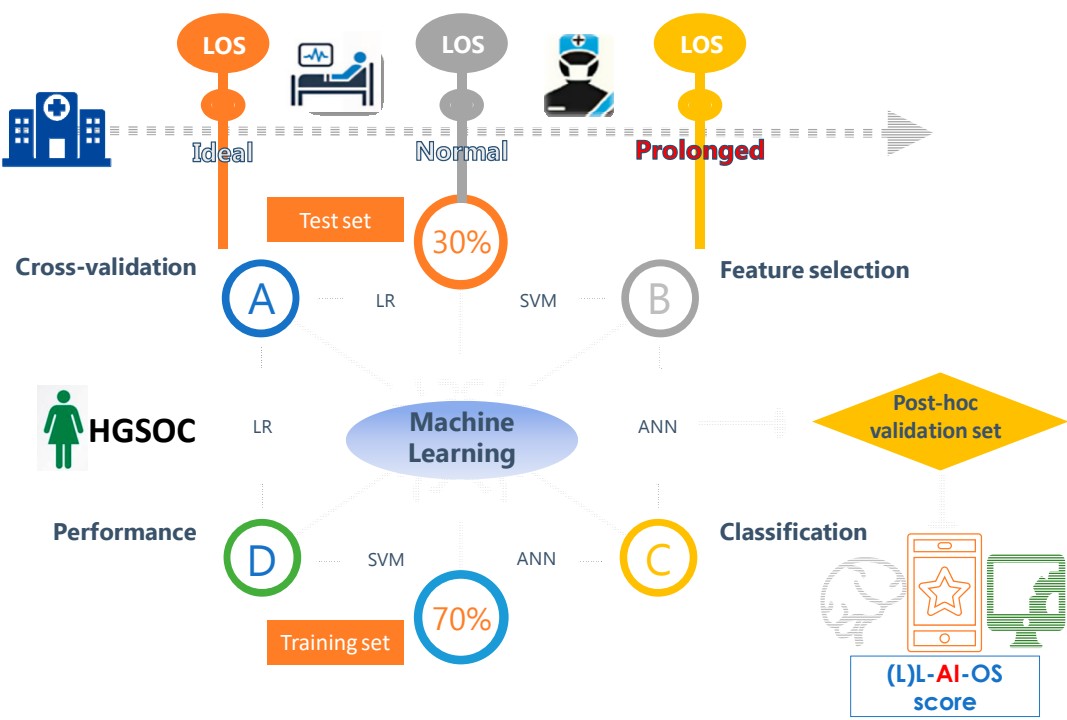

**Figure 1.** Workflow and organization for model selection, internal and post-hoc validation of the length of stay (LOS) prediction in advanced stage high grade serous ovarian cancer patients (HGSOC) following their cytoreductive surgery using Machine Learning. Both classification and numerical LOS predictions were interrogated.

**Table 2.** Cohort descriptive statistics.

| Variable | Age (Years) | Surgical Complexity Score (SCS) | Disease Score (DS) | Pre Surgery Alb (ALB) | Estimated Blood Loss (EBL) (mL) | Operative Time (OT) (min) | LOS (Days) |
|---|---|---|---|---|---|---|---|
| Mean | 64 | 4 | 2 | 39 | 484 | 181 | 6 |
| Standard Deviation | 10 | 2 | 1 | 4 | 411 | 76 | 4 |
| Minimum | 41 | 2 | 1 | 27 | 100 | 65 | 3 |
| Maximum | 90 | 11 | 3 | 49 | 4000 | 480 | 24 |
| Tenth Percentile | 50 | 2 | 2 | 34 | 200 | 105 | 4 |
| Lower Quartile | 56 | 2 | 2 | 36 | 250 | 120 | 5 |
| Median | 65 | 3 | 2 | 38 | 400 | 160 | 5 |
| Upper Quartile | 73 | 4 | 2 | 41 | 500 | 225 | 7 |
| 90th centile | 77 | 6 | 3 | 43 | 900 | 285 | 9 |

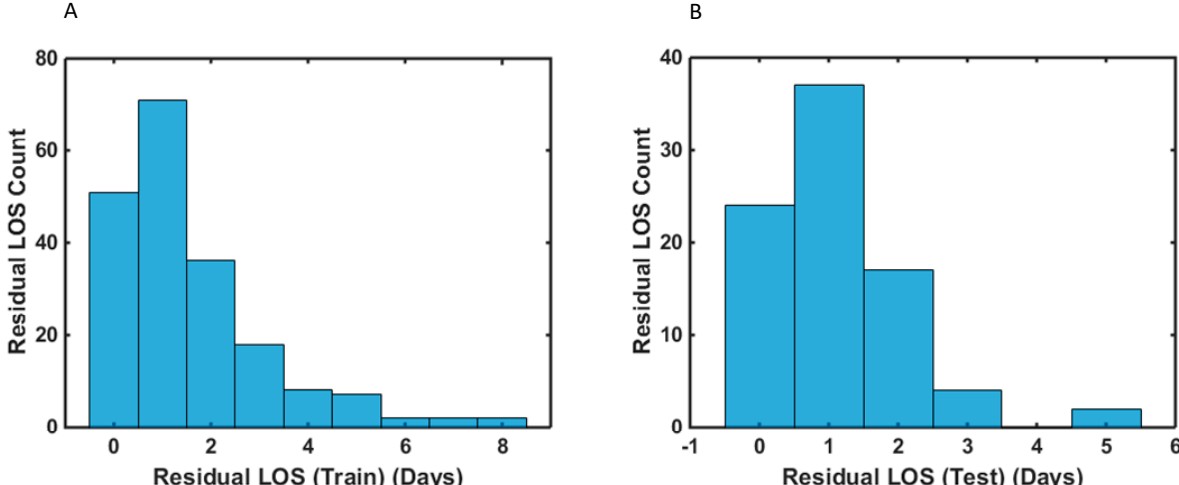

**Figure 2.** Histograms of the residual length of stay (LOS) distribution of advanced-stage HGSOC patients following their cytoreductive surgery for the: (**A**) training (**B**) test samples. Residual LOS is the difference between the observed LOS values and the model-predicted LOS values.

Three well-defined time points were interrogated; namely interval 1, which differentiated patients with ideal LOS ($\leq$five days) from patients who stayed $\geq$five days; interval 2, which differentiated patients with prolonged LOS $\geq$ seven days (Definition 1); and interval 3, which differentiated patients with prolonged LOS $\geq$ 10 days (Definition 2). Results from binary classification predictions between patients with ideal LOS (Interval 1) vs. non-ideal LOS reached the mark of 70% accuracy in the validation set, best estimated by LR. For the prediction of prolonged LOS (Definition 1; $\geq$7 days), ANN outperformed SVM and LR reaching an accuracy of 76%. For prolonged LOS (Definition 2; >90th centile), the best accuracy was achieved by LR, followed by SVM (98% vs. 94%, respectively). However, as the F-score is widely used in ML as a test of accuracy, for intervals 1 and 2, ANN proved to be more effective as a classification algorithm. For interval 3, SVM proved to be the best model. Herein, the F-Score figure of merit as a choice criterion was 63%, 59%, and 66% for intervals 1, 2, and 3, respectively (Table 3). Notably, 0% values in the sensitivity measures, F-score, and G-score for interval 3 were observed. Indeed, in the test set, the LR model misclassified all the samples from the training set, which was actually referring to the group of patients with LOS < nine days, thus generating a sensitivity of 0%. Subsequently, this result automatically generated values of 0% for F-score and G-score, values Feature selection identified SCS, pre-surgery albumin, EBL, OT, bowel resection with stoma formation, and severe postoperative complications (CD3–5) as statistically significant. The correlation structure of the continuous and categorical variables was demonstrated by a correlation heatmap, as shown in Figure 3.

**Table 3.** Prediction performance of the three models for (1) ideal (2) prolonged 1; (above median) (3) prolonged 2; (above 90th centile) length of stay (LOS).

| LOS (1 ≤ 5 d)—Ideal (2 ≥ 6 d)—Prolonged 1 (3 ≥ 9 d)—Prolonged 2 | Model | Set | Accuracy | Sensitivity | Specificity | F-Score | G-Score |
|---|---|---|---|---|---|---|---|
| 1 | ANN | TRAIN | 93% | 93% | 93% | 93% | 93% |
| 1 | ANN | CV | 62% | 62% | 62% | 62% | 62% |
| 1 | ANN | TEST | 64% | 67% | 59% | 63% | 63% |
| 1 | SVM | TRAIN | 79% | 91% | 68% | 78% | 79% |
| 1 | SVM | CV | 72% | 79% | 65% | 71% | 72% |
| 1 | SVM | TEST | 69% | 89% | 45% | 60% | 63% |
| 1 | LR | TRAIN | 73% | 78% | 68% | 73% | 73% |
| 1 | LR | TEST | 70% | 91% | 45% | 60% | 64% |
| 2 | ANN | TRAIN | 98% | 99% | 98% | 98% | 98% |
| 2 | ANN | CV | 71% | 64% | 76% | 69% | 70% |
| 2 | ANN | TEST | 76% | 45% | 87% | 59% | 63% |
| 2 | SVM | TRAIN | 100% | 100% | 100% | 100% | 100% |
| 2 | SVM | CV | 72% | 67% | 75% | 71% | 71% |
| 2 | SVM | TEST | 68% | 39% | 79% | 52% | 56% |
| 2 | LR | TRAIN | 75% | 51% | 90% | 65% | 68% |
| 2 | LR | TEST | 75% | 35% | 90% | 50% | 56% |
| 3 | ANN | TRAIN | 97% | 96% | 97% | 96% | 96% |
| 3 | ANN | CV | 80% | 54% | 84% | 66% | 67% |
| 3 | ANN | TEST | 89% | 50% | 90% | 64% | 67% |
| 3 | SVM | TRAIN | 97% | 81% | 100% | 90% | 90% |
| 3 | SVM | CV | 86% | 35% | 94% | 51% | 57% |
| 3 | SVM | TEST | 94% | 50% | 95% | 66% | 69% |
| 3 | LR | TRAIN | 90% | 46% | 97% | 63% | 67% |
| 3 | LR | TEST | 98% | 0% | 100% | 0% | 0% |

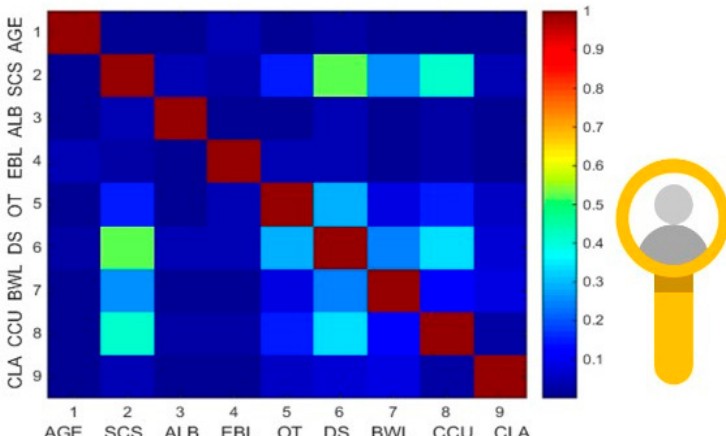

**Figure 3.** Correlation heatmap showing the pairwise associations between the variables selected for the calibration model (significant variables with *p* < 0.05). The association between the continuous variables was examined using Pearson's correlation. The highest correlation was observed between SCS and DS. SCS = Surgical Complexity Score; ALB = Pre-Surgery Albumin; EBL = Estimated blood loss; OT = Operative time; DS = Disease Score; BWL = Bowel Resection; CCU = CCU Admission; CLA = Clavien Dindo complication 3–5.

To promote clinical implementation, a user-friendly, password-protected GUI was developed (the Leeds-LAIOS score) to enable standardization of postoperative LOS prediction (as a continuous variable) in HGSOC women following their cytoreductive surgery. The interface procures probability results and visualization of the risk position for LOS for clinical implementation and early interpretation by surgeons. The statistical parameters for the model calibration were as follows:

RMSE (Calibration): 2.2281; RMSE (Cross-Validation): 2.5846; RMSE (Prediction): 1.5079; Bias (Calibration): 0.0102; Bias (Cross-Validation): 0.0000; Bias (Prediction): 0.119. These values demonstrated the suitability of the regression model to be based on neural networks. Hence, the RMSE was a good estimator for the standard deviation of the LOS distribution by ±two days. To validate the regression model, residual plots visually confirmed the validity of the model (Figure 4). For the numerical prediction (LOS as a continuous variable), a difference between the measured and expected value of up to 2 days was estimated in 93% of the samples; a difference of up to 1 day in 73% of the samples, and finally, in 30% of the samples the LOS prediction was equal to that measured clinically on the patients.

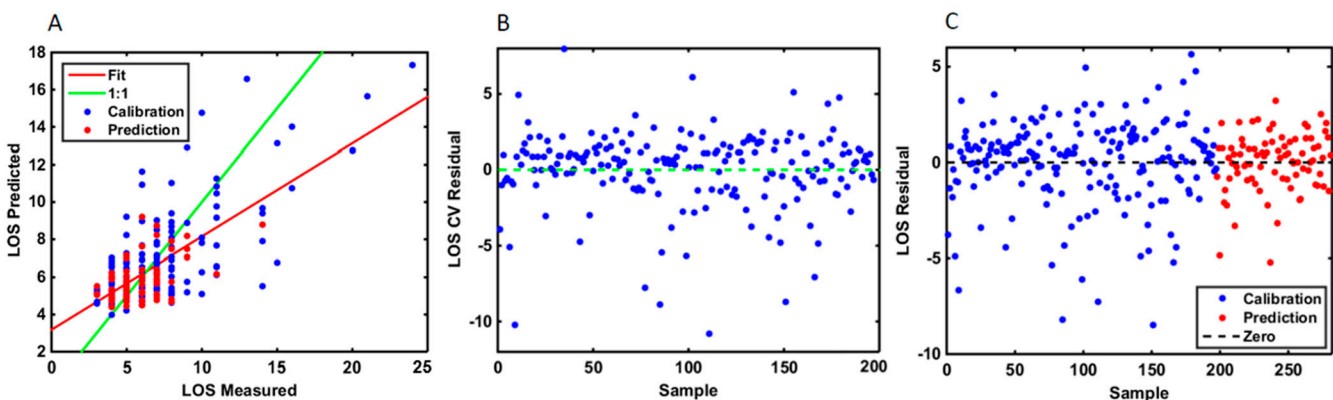

**Figure 4.** (**A**) Comparative curve analysis showing the measured and the predicted values for the calibration and prediction samples using the LOS as a continuous variable. The plot suggests that the ANN model had the optimal calibration power for LOS prediction. (**B**) Residual plot for LOS using

predicted samples in cross-validation methodology. A residual is a measure of how far away a point is vertically from the regression line. Simply, it is the error between a predicted value and the observed actual value. The residual plot has the residual values on the *y*-axis and the independent variable on the *x*-axis. A good residual plot has a high density of points close to the origin and a low density of points away from the origin. It is symmetric about the origin showing the residual errors are approximately distributed in the same manner. (**C**) Residual plot of the differences between the observed data values and the ANN model predicted values for the calibration and the prediction samples.

## 4. Discussion

To the best of our knowledge, this study is the first to construct optimal data-driven ML/DL models to accurately predict and comprehensively quantify postoperative LOS in HGSOC patients. Such predictive models have already tested the intra-institutional performances, and shown to improve surgical morbidity and mortality by 45% and 31%, respectively [30]. Our approach allowed for the comparison of selected ML algorithms to identify the method with the most favorable performance for predicting LOS at various time points. Different definitions for prolonged LOS were interrogated for the differential binary classification predictions. A definition of prolonged LOS as longer than the median LOS may help models achieve better generalizability. Not surprisingly, conventional regression performance was comparable with ML/DL performance, likely due to class imbalance and varying degrees of collinearity. Arguably, LR was easy to implement, required the least computing power, and would not carry the "black box" feature common to many ML models.

We demonstrated a direct clinical application of ML for the numerical LOS prediction allowing for comparison against the more conventional (LR) approach. We developed a GUI, which can be used to address one of the main patient's concerns; that is, the duration of hospital stays, using readily available clinical data from the hospital environment. The numerical models focused on mining different LOS patterns for patients with different LOS (even 1-day apart) and the modeling successfully discriminated between patients with different LOS patterns. At first, the models were evaluated using all available clinical variables. However, the initial results were not significant prompting further refinements to produce clinically meaningful outputs. The internally validated performances were satisfactory with good calibration. The calibration curves visually showed that the averaged predicted LOS probability of the models were consistent with the observed outcomes across different LOS risk groups from low to high. By demonstrating the utility of ML algorithms, we determined the impact of several variables, shown to be predictors of short-term outcomes in EOC patients [14,31].

By using the F-score as a choice criterion, ANN proved to be more effective as a classification algorithm except for LOS > nine days prediction, whereas SVM proved to be the best model. This varying accuracy could be due to class imbalance, albeit it was not our scope to show that AI outplays conventional methods. Nevertheless, this versatility can be useful and help rationalize the transition from conventional statistics, which clinicians are mostly familiar with, toward the wider application of AI frameworks. To overcome the resilience for the widest adaptation in the clinical environment, Explainability Artificial Intelligence (XAI) can be powerful to unveil the potential "black box" of AI [32]. Our team has pioneered the implementation of XAI in the EOC trajectory and provided insight into the potential influence of human factors on surgical decision-making at cytoreduction [33,34]. Due to some features' collinearity, it was also inevitable that a post-operative feature was included amongst the list of features forming the post-hoc model, which would have even affected the overall model's accuracy.

Our study shows that it is possible to base the assessment of a unit's performance on LOS because it adjusts for the case mix and complexity of the operation. We identified an association between longer LOS and several clinical factors following HGSOC surgical cytoreduction, many of which can be potentially affected by surgical decisions. Women with an HGSOC diagnosis are generally older, have additional comorbidities, and present with disseminated disease. The advanced-stage disease is commonly associated with

weight loss, severe malnutrition, and impaired gastrointestinal function, resulting in a protracted surgical recovery [35]. Primary chemotherapy could potentially impact some of these factors, as shown to decrease intraoperative blood loss, earlier resumption to ambulation, and return of intestinal function [36]. Nevertheless, in our study, the timing of surgery did not impact hospital stay.

Identification of at-risk patients for a prolonged hospital stay may aid in targeted interventions to reduce hospital stay, improve the quality of care, and decrease healthcare costs. The novel (L)L-AI-OS scoring system was developed to standardize the LOS prediction using an AI algorithm, which is particularly helpful when counseling EOC patients about their peri-operative risks. It mostly incorporates pre-operative and intra-operative parameters, and can be used in clinical trials, internal audits, routine practice, and future benchmarking (Figure 5). Inevitably, postoperative complications were included as an independent variable in the scoring system, because for modeling the extreme LOS, incorporation of data from post-operative events is essential affecting 50–70% of patients. In this respect, complication rates may be reported separately as they emphasize severity. Therefore, the score can serve as a basis for clinical recommendations to mitigate the risk of long hospitalization by minimizing the risk of postoperative complications in addition to serving as a proxy for patient acuity.

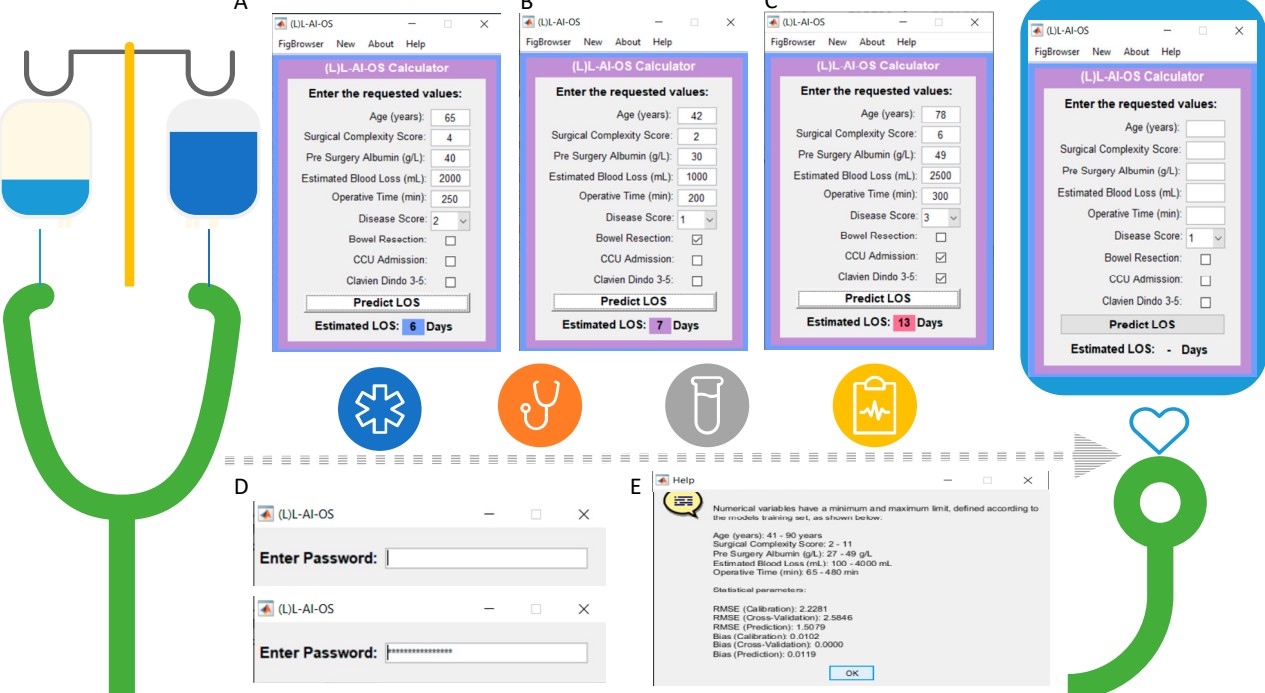

**Figure 5.** Graphical User Interface (GUI) presentation of the Leeds L-AI-OS score. (**A–C**): The (L)-LAIOS button navigates to the feeder interface. The length of stay (LOS) index assigns weighted risk points to each significant factor in a scoring system to predict hospital stay (**D**) The GUI is password protected. (**E**) The help button routes to more information about the prediction model including the values of the numerical variables and the statistical parameters.

Despite an ERAS pathway being employed in cytoreductive surgery, the operative factors remain a barrier to early discharge, yet differ according to the complexity of the surgery. Operative time is an often-analyzed factor influencing outcomes of patients undergoing cytoreductive surgery. It procures a surrogate for the technical complexity of a procedure. Surgical complexity is associated with prolonged LOS even with an established ERAS pathway in place [31]. Intrinsic non-modifiable predictors for LOS differ with operative complexity, and this should be considered when planning benchmarking and research across units. Our mean LOS was comparable to the UK ERAS data, which includes

the National Bowel Cancer Audit Programme [35]. The length of hospital stay is also known to be proportional to compliance [36]. A combination of worse compliance, increased morbidity, and slower functional recovery due to the case-mix could be responsible for the delayed LOS. The delayed recovery pattern fits with the literature demonstrating that early postoperative deviation from ERAS goals and compliance could predict prolonged LOS [37].

All the cytoreductive surgeries were elective. Not surprisingly, operative time reflecting greater surgical effort, potentially increasing the surgical risk would predispose to longer recovery times [10]. Significant factors found in our study could be used to formulate individual disease-specific treatment pathways and early discharge planning to decrease inpatient LOS. Additionally, prioritizing laboratory tests and avoiding test duplication can effectively decrease the LOS [38]. Nevertheless, Gynaecologic Oncology specific prevention bundles did not affect LOS or readmission rates, albeit, they significantly reduced wound infection rates [39].

Furthermore, an individual prolonged LOS risk profile can be used as a decision-making aid to the physician's subjective judgment while adjusting a patient's LOS [40]. Advanced HGSOC patients frequently require extensive procedures including bowel resections and upper abdominal surgery. It is expected that patients requiring stomas will stay longer compared to those with continuity of their bowels maintained. Nonetheless, patients with the longest delay to the initial stoma nurse visit had the longest LOS [41]. The timing of performing such operations early during the week may reduce LOS due to the absence of stoma teaching over the weekend.

An inverse relation between serum albumin and LOS among patients with gynecological cancers has been documented [42]. This is likely due to increased postoperative complications [43,44], reflecting adverse survival outcomes [45]. The more severely malnourished the patient is, the lower the levels of serum albumin. Addressing malnutrition and poor quality of life may decrease patient length of hospitalization and hospital readmission [46]. The key recommendations against cancer-related malnutrition have been recently updated [47]. Nutritional supplementation may produce a small weight gain benefit in the elderly group but does not provide evidence of improvement in functional benefit or reduction in LOS [48]. Meaningful reductions in SSIs can be achieved by implementing a multidisciplinary care bundle at a hospital-wide level, resulting in bigger differences in wound-related rather than organ-space SSIs [39]. Prognostic models of SSIs using daily clinical wound assessment have been reported [49]. As morbidity-related infections and wound complications from surgery can increase hospital stay, nutrition support during the peri-operative period is warranted [13]. A sensible approach would be to implement oncology protocols providing recommendations for nutritional screening, assessment tools, and supplementations. A previous study found that gynecological cancer patients with two or more pre-existing co-morbidities had significantly longer LOS than those with one or no co-morbidities [50]. Notably, we failed to show any effect of patient obesity on LOS, but an indirect association cannot be excluded, obesity being the culprit for increased OT and SSIs [51].

Large ERAS studies have also demonstrated that stoma formation when performed as part of the primary colorectal surgery prolongs LOS. Despite preoperative stoma education, it was therefore not surprising that in our study, stoma formation remained consistently an independent predictor of prolonged LOS. Our study also shows that over the four-year period, the overall median LOS has significantly reduced. This was partly due to the increased proportion of patients experiencing ideal LOS, which was promoted as a good care measure following the introduction of the ERAS pathway in our institution [27]. Implementation of ERAS pathways for advanced-stage EOC results in a shorter LOS due to earlier recovery and a lower rate of readmission, with no increase in morbidity or mortality [3]. From other short-term outcomes, we only assessed the rate of hospital readmission. There were no rapid postoperative deaths -a negative outcome- which would positively influence LOS. Occasionally, a patient who is rapidly discharged can be rapidly readmitted if not successfully recovered.

Prolonged LOS, CCU stays, hospital readmissions, and aggressive therapies, such as chemotherapy and surgery have all come under scrutiny due to emphasis given on improved palliative care and quality of life for patients near their end of life. Critical care admission was an independent predictor of LOS. This is in line with other studies [31], and is likely a reflection of pre-operative frailty albeit co-morbidities had no effect on the cohort. A large proportion of our CCU admissions were elective, which suggests potential over-utilization, yet it remains practically difficult to achieve the ERAS goals in a critical care ward [52].

The LOS models for this group of patients are relatively new in the UK. The American College of Surgeons' National Surgical Quality Improvement Program (NSQIP) risk calculator is an alternative model that predicts LOS based on preoperative data. It appears that in the EOC population, it significantly underestimates the LOS probably because surgical coding does not include several sub-procedures during cytoreductive surgery [53].

This was a single-center retrospective study with a limited rate of heterogeneity in the study population, which may differ from other tertiary unit settings. The study suffers a small risk for "time-dependent" bias, as CD3–5 complications were a time-varying feature introduced after patients had already spent some time in the hospital. This can be a problem with regression models albeit, in this study, we explored feature associations rather than causative inferences. Nevertheless, ML retains the strength of the structural model used for the prediction even when applied in other populations and reveals different prediction features. We noted the recording of the discharge date as a single endpoint. We did not record the "medically fit for discharge" date to account for whether the discharge was delayed for social reasons. Studies show that having a dedicated discharge coordinator with effective early discharge planning following four days of inpatient care can significantly reduce inpatient LOS [54]. The length of stay can be longer for patients discharged to a nursing home or rehabilitation facility [55]. Audits designed to identify stays that are longer than expected for reasons other than surgical performance will potentially support the development of better discharge pathways.

This information can assist the hospital trust with discharging patients within the ideal length of time and facilitate resource allocation for the delivery of care. Discharging patients earlier, not only decreases costs, but also the risk of patients developing hospital-acquired infections. We anticipate the formation of collaboration net to allow comparison between the trusts, and to identify potential 'outliers'. In this way, centers with poorer outcomes may improve their care through the comparison of practices driven by a competitive spirit [56]. Validation of our results in prospective studies will valuably integrate AI methods in clinical practice and benefit clinicians and cancer patients.

## 5. Conclusions

Length of stay is a measurable outcome that can be used as a benchmark of surgical care. We demonstrated the development and application of both quantitative and qualitative models to predict LOS in advanced-stage EOC patients following their cytoreduction. These predictive ML algorithms may facilitate the quality improvement of modern care by enhancing prediction accuracy for LOS. Complex EOC cytoreduction may be associated with a rise in postoperative LOS, a cost-related outcome. Using ML methods, we more accurately refined potentially modifiable factors delaying hospital discharge, which may further inform services performing root cause analysis of LOS. For this inherently high-risk population, our prediction scoring system serves as critical information when counseling patients about the peri-operative risks.

**Author Contributions:** Conceptualization, A.L.; methodology, A.L. and D.L.D.D.F.; software, A.L. and D.L.D.D.F.; validation, D.L.D.D.F. and E.K.; formal analysis, A.L., Y.S.T., A.Z. and E.K.; investigation, A.L. and R.J.; resources, G.S. and S.M.; data curation, A.L. and Y.S.T.; writing—original draft preparation, A.L.; writing—review and editing, A.T., R.J., R.H., T.B., D.N., E.K., G.T., K.M.G.d.L. and D.D.J.; visualization, A.L. and D.L.D.D.F.; supervision, D.D.J.; project administration, T.B. and G.S. All authors have read and agreed to the published version of the manuscript.

**Funding:** This research received no external funding.

**Institutional Review Board Statement:** The study was conducted in accordance with the Declaration of Helsinki, and approved by the Leeds Teaching Hospitals Ethics Committee of St James's Institute of Oncology (MO20/133163/18.06.20).

**Informed Consent Statement:** Informed consent was obtained from all subjects involved in the study.

**Data Availability Statement:** The data presented in the manuscript are available upon request by the corresponding author.

**Acknowledgments:** We would like to thank all the members of the Gynaecological Oncology team at St James's University Hospital, Leeds who managed the patients enrolled in the study. All team members have consented to this acknowledgement.

**Conflicts of Interest:** The authors declare no conflict of interest.

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
