# Peer review of "Stratification of Length of Stay Prediction following Surgical Cytoreduction in Advanced High-Grade Serous Ovarian Cancer Patients Using Artificial Intelligence; the Leeds L-AI-OS Score"

_curroncol, doi:10.3390/curroncol29120711_

Round 1

Reviewer 1 Report

The manuscript develops the concept of using AI algorithms to predict LOS as a binary outcome for different time-points. This remains an important topic in gynecological oncology literature because it can be used as a benchmark of short-term care, and most info is derived from the surgical oncology literature. However, testing LOS as a continuous outcome makes it even more interesting and certainly represents a novel approach by performing a post-hoc validation for clinical feasibility.

The abstract summary and intro are clearly presented, explaining the rationale for the study. I have no concerns about methodology. In terms of the results, the model accuracies are satisfactory but the authors need to explain why different ML algorithms champion the prediction accuracies for different outcomes. Did they attempt to prove whether ML outperforms logistic regression to enhance use of more clinically transferable algorithms? It would have also been ideal if the post-hoc model included only pre- and peri-operative variables; If I was a patient, I would still need to know exactly how many days I will stay in the hospital, but it is inevitable to include the post-operative complications as there is an obvious inter-relation. The discussion is very well written, and I agree with the strongholds and the potential study limitations. How will clinicians become less resilient in the adoption of these AI methodologies?

Author Response

Please the attachement

Reviewer 2 Report

The author should be more concerned about these comments.

  • The author used different words in the current manuscript, such as artificial intelligence, machine learning, and deep learning. However, the author used an artificial neural network (ANN) and conventional logistic regression to predict continuous and 25 binary LOS outcomes for HGSOC patients. In my view, the proposed method is not a deep learning method. Could the author design not to use the word deep learning in the manuscript? 
  • As the feature selection, the author used one by 139 one against the continuous vector of LOS classification responses. There are many feature selection methods. Could the author describe why the author uses this method? 
  • Could the author present the concrete contribution of this research?
  • Three machine learning methods were used as the classifier, including LR, SVM, and MLP. However, the theory of these machine learning techniques is needed.
  • The 16 features (variables) were used in the classification process after applying the feature selection method, as shown in Table 1. Could the author present all variables? The audience could see the original features and the selected features.
  • Does the author evaluate the proposed method using the full features? The author should report the result. So the audience could understand that the feature selection shows a significant result.
  • As shown in Table 2, the author reported the experimental results on the train, CV, and test sets. Please describe how to evaluate these three sets. The author also evaluated using a CV. Please report the standard deviation value. What is the full name of the CV? Is it a cross-validation method?
  • As shown in Table 2, in the last row, could the author explain why the sensitivity, F-score, and G-score values are 0%? Which machine learning technique shows the best prediction? Please make it in bold font.
  • Up-to-date references are required.

Round 2

Reviewer 2 Report

Thank you, the author, for responsed all the concerns points. The revised manuscript is ready to be published.